# Reproductive Performance Following Hysteroscopic Surgery for Uterine Septum: Results from a Single Surgeon Data

**DOI:** 10.3390/jcm10010130

**Published:** 2021-01-02

**Authors:** Ertan Saridogan, Mona Salman, Lerzan Sinem Direk, Ali Alchami

**Affiliations:** Elizabeth Garrett Anderson Institute for Women’s Health, University College London Hospital, 250 Euston Road, London NW1 2PG, UK; mona.moghazy@nhs.net (M.S.); zchalsd@ucl.ac.uk (L.S.D.); ali.alchami@crgh.co.uk (A.A.)

**Keywords:** uterine septum, septum incision, pregnancy, live birth, infertility, miscarriage

## Abstract

Uterine septum can negatively affect reproductive outcomes in women. Based on evidence from retrospective observational studies, hysteroscopic incision has been considered a solution to improve reproductive performance, however there has been recent controversy on the need for surgery for uterine septum. High quality evidence from prospective studies is still lacking, and until it is available, experts are encouraged to publish their data. We are therefore presenting our data that involves analysis of the patient characteristics, surgical approach and long-term reproductive outcomes of women who received treatment for uterine septum under the care of a single surgeon. This includes all women (99) who underwent hysteroscopic surgery for uterine septum between January 2001 and December 2019. Of those 99 women treated for intrauterine septum who were trying to conceive, 91.4% (64/70) achieved pregnancy, 78.6% (55/70) had live births and 8.6% (6/70) had miscarriages. No statistically significant difference was found in the live birth rates when data was analyzed in subgroups based on age, reason for referral/aetiology and severity of pathology. Our study results support the view that surgical treatment of uterine septa is beneficial in improving reproductive outcomes.

## 1. Introduction

Septate uterus is the most common congenital uterine anomaly, making up about 55% of all cases [1]. Its estimated prevalence is 0.2–2.3% in women of reproductive age [2]. Septate uterus is caused by the failure of resorption of the uterovaginal septum at nine weeks gestation after the Mullerian duct fusion during the development of the female genital tract [3]. This may be complete or partial failure of resorption, leading to either a complete septum or subseptate uterus [3]. There is no universally accepted classification system of congenital uterine anomalies including septa, however the most commonly used are the European Society of Human Reproduction and Embryology/European Society for Gynaecological Endoscopy (ESHRE/ESGE) classification system, the American Society of Reproductive Medicine (ASRM) system, and the Congenital Uterine Malformation by Experts (CUME) criteria, which allow objective classifications based on 3D ultrasound [4,5,6,7]. Women with septate uterus have an increased risk of subfertility and pregnancy complications including pregnancy loss, preterm delivery, fetal mal-presentation and intrauterine growth retardation, resulting in poor outcomes including higher perinatal morbidity and mortality [2,8]. A recent systematic review by Rikken et al., commented that although the biological basis of the unfavorable reproductive outcome with uterine septa is yet to be proven, the gross anatomy of the septum or histological difference in the endometrium covering the septum or gene expression as lower expression of HOXA10 genes and VEGF receptor genes could possibly attribute for the impaired reproductive outcome [9].

Hysteroscopic septum incision guided by preoperative three- dimensional ultrasound imaging is considered a solution and has been standard treatment for those patients with impaired reproductive outcomes. Although high quality evidence through randomized controlled prospective trials is still lacking on the efficacy of surgical treatment, the current evidence from retrospective observational studies and non-randomized comparative studies shows that septum surgery is beneficial in symptomatic women [7,8,10,11,12,13,14]. There has been a recent controversy on the need for surgery for septum [15]. In their retrospective study, Rikken et al. reported that septum resection did not improve the reproductive outcomes and that compared to expectant management this led to decreased chances of an ongoing pregnancy. The study was criticized due to bias and several limitations [13,14,16,17].

Such controversy should encourage clinicians to publish their data in such fields until prospective studies are planned and in place and evidence is robust. We are therefore presenting our data that involves analysis of the patient characteristics, surgical approach and long-term reproductive outcomes of women who received treatment for uterine septum under the care of a single surgeon.

## 2. Methodology

### **2.1.** Study Design and Population

This was a retrospective review of consecutive cases who underwent surgical treatment for uterine septum by a single operator (ES) during the period between January 2001 and December 2019. Patients were identified from clinical notes, consultant diaries and hospital operating theatre records. Each patient’s records were checked individually and data was collected on patient demographics, presenting symptoms, number and outcome of previous pregnancies, pre-operative diagnostic methods and their findings, operative details and complications and post-operative reproductive outcomes. In our center, diagnosis of septum was usually made by 3D ultrasound using criteria as described by Salim et al. [18]. When the septum reached the internal os it was described as a complete septum, and a partial septum (or subseptate uterus) was diagnosed when the septum did not reach the internal os. Some patients were referred for surgery after diagnosis with MRI, hysteroscopy, hysterosalpingography or a combination of hysteroscopy and laparoscopy. For each patient, relevant data was procured from their electronic or paper records or clinic letters. Findings were entered into a password-protected Excel spreadsheet (Microsoft Corporation, Seattle, WA, USA) with adherence to standards of good clinical practice in research to avoid breaching patient confidentiality.

All procedures were performed by a single surgeon (ES) under general anesthesia as a day procedure. Hysteroscopic procedures were performed using a mini-hysteroscope (Alphascope 3.5 mm with a 0° 1.9 mm fused fiber optic, Johnson and Johnson, or Bettochi operating hysteroscope 4.2 mm with a 30° 2.9 mm rod lens optic, Karl Storz) with inflow, outflow and 5 or 7 French operating channels. 2D ultrasound guidance was utilized for the majority of patients. Concomitant laparoscopy was not routinely used, unless there was a finding of partial bicornuate uterus or it was indicated for other reasons such as adnexal mass, pelvic pain or endometriosis. Intrauterine septa were divided using hysteroscopic scissors and/or 5 French bipolar electrodes until a regular cavity with a fundal myometrial thickness of 11–12 mm was achieved. A Copper T IUCD was inserted into the cavity and a repeat hysteroscopy was performed to confirm its correct positioning. The IUCD was removed 6–8 weeks later. Further hysteroscopies or ultrasound examinations were not routinely performed, unless clinically indicated.

### 2.2. Statistical Analysis

Statistical analysis was performed using Excel (Microsoft Corporation, Seattle, WA, USA). The demographic characteristics of included patients are expressed as median (range), and proportions are expressed as percentages with 95% confidence intervals (CIs). Subgroup data has been analyzed using the Chi-squared test in order to give a *p*-value to ascertain whether the data is of statistical significance.

### 2.3. Ethical Approval

The study was assessed by the NHS Health Research Authority ‘Defining Research’ decision tool and full ethical review by an NHS Research Ethics Committee or NHS/Health and Social Care Research and Development office was not required as this study was considered a service evaluation (www.hra-decisiontools.org.uk/research/).

## 3. Results

During the study period, 99 women were treated for intrauterine septum, with a mean age of 35.6 years (range 24–48 years). The reasons for referral are shown in Table 1. The majority of patients was referred with a history of infertility, followed by miscarriage (1 or 2 miscarriages) and recurrent miscarriage (≥3 miscarriages). The total number of miscarriages in the recurrent miscarriage group was 51, with an average of 3.6 miscarriages (range 3–5).

Preoperative diagnosis was made using 3D ultrasound (with or without saline infusion) in 80 of 99 (80.8%; 95% CI, 66.9–94.7) cases, ultrasound and MRI in 6 of 99 (6.1%; 95% CI, 1.4–10.8) cases, hysteroscopy in 5 of 99 (5.1%; 95% CI, 0.7–9.4) cases, ultrasound and hysterosalpingography in 2 of 99 (2.0%; 95% CI, 0–4.8) cases, MRI only in 1 of 99 (1.0%; 95% CI, 0–3.0) cases, hysteroscopy and laparoscopy in 2 of 99 (2.0%; 95% CI, 0–4.8) cases, and a combination of ultrasound, hysterosalpingography and MRI in 1 of 99 (1.0%; 95% CI, 0–0.3) cases. In 2 of 99 (2.0%; 95% CI, 0–4.8) cases the method of diagnosis is unknown. 30 of 99 (30.3%; 95% CI, 21.3–39.4) patients had a complete septum, 62 of 99 (62.6%; 95% CI, 53.1–72.2) had subseptate uteri and 5 of 99 (5.1%; 95% CI, 0.7–9.4) had a residual septum only (following a previous septum operation elsewhere). In two cases (2.0%; 95% CI, 0–4.8) the degrees of uterine septum were not found in available records.

Following treatment, 21 of 99 (21.2%; 95% CI 13.2–29.3) women were lost to follow-up, 5 of 99 (5.1%; 95% CI, 0.7–9.4) are not trying for pregnancy yet, 1 of 99 (1.0%; 95% CI, 0–3.0) is currently awaiting IVF and 1 of 99 (1.0%; 95% CI, 0–3.0) decided against fertility treatment. One woman passed away due to unrelated causes. The remaining 70 of 99 patients have been analyzed for reproductive outcomes. Overall, of those women are known to be trying to conceive, 64 of 70 (91.4%; 95% CI, 84.9–98.0) achieved pregnancy, 6 of 70 (8.6%; 95% CI, 2.0–15.1) had miscarriages (total 11 pregnancy losses) and 55 of 70 (78.6%; 95% CI, 69.0–88.2) had live births. 37 of 70 (52.9%; 95% CI, 41.2–64.6) patients conceived spontaneously or with IUI (in 2 cases). Of these women, 33 of 37 (89.2%; 95% CI, 79.2–99.2) had live births, 3 of 37 (8.1%; 95% CI, 0–16.9) had miscarriages and 1 of 37 (2.7%; 95% CI, 0–7.9) women had an unknown pregnancy outcome. 27 of 70 (38.6%; 95% CI, 27.2–50.0) patients conceived with IVF. Of these women, 22 of 27 (81.5%; 95% CI, 66.8–96.1) had live births, 3 of 27 (11.1%; 95% CI, 0–23.0) had miscarriages and 1 of 27 (3.7%; 95% CI, 0–10.8) women had an unknown pregnancy outcome. One woman (3.7%; 95% CI, 0–10.8) had a termination in the 2nd trimester for a chromosomal abnormality. 6 of 70 (8.6%; 95% CI, 2.0–15.1) women received IVF treatment but failed to conceive. The post-treatment reproductive outcomes are summarized in Table 2.

### Live Birth Rates in Subgroups

Data was analyzed in subgroups based on age, reason for referral and degree of intrauterine septum. Live births occurred in 8 of 13 (61.5%; 95% CI, 35.1–88.0) cases in the ≥40 age group, in 42 of 51 (82.4%; 95% CI, 71.9–92.8) in the 30–39 age group and in 5 of 6 (83.3%; 95% CI, 53.5–100) in the <30 age group. The differences in live birth rates across the age groups were not statistically significant (*p* = 0.25) (Figure 1). IVF pregnancies were more common in the ≥40 age group compared to younger age women, but this was again statistically non-significant.

The live birth rates were not statistically different in infertility, recurrent miscarriage and miscarriage groups (*p*= 0.62) (Table 3).

14 out of 18 (77.8%, 95% CI, 58.6–97.0) women with a complete septum, 36 out of 47 (76.6%, 95% CI, 64.5–88.7) women with a partial septum and 3 out of 4 (75.0%, 95 CI, 32.6–100) women with a residual septum had a live birth (*p* = 0.99).

## 4. Discussion

Our data shows improved reproductive outcomes for patients with treated uterine septa; 91.4% of women achieved pregnancy; 78.6% had live births and only 8.6% had miscarriages. A review by Homer et al. pooled live birth rates and pregnancy loss rates of 16 studies; the live birth rates before and after septum resection were 3% versus 80% and the rate of pregnancy loss was 88% versus 14% respectively [19]. Our live birth rates are very similar to the figure in this review in a group of women, the majority of whom presented with a history of poor reproductive performance.

Our data has shown similar live birth rates in women presenting with recurrent miscarriage and infertility (80.0% and 78.9%, respectively). This is in contrast to some of the published literature that has shown that women presenting with miscarriage have a higher live birth rate following hysteroscopic metroplasty than women presenting with infertility [20,21]. Lower pregnancy rates in women with a history of infertility are not surprising, as they are more likely to have additional causes for their infertility, whereas women with recurrent miscarriages may be more likely to have a successful obstetric outcome once the septum has been treated. A recent systematic review on the pathophysiology of the septate uterus included 38 studies and concluded that there is no clear biological basis for the unfavorable reproductive outcome in women with uterine septa, however factors such as the gross anatomy of the septum or histological difference in the endometrium covering the septum or gene expression could attribute for the impaired reproductive outcome. Large studies are needed to confirm clinical relevance [9]. Infertile women in our group had access to IVF after the treatment of their septum and this probably explains how their pregnancy rates reached similar levels to those who presented with miscarriages.

We also found similar live birth rates after treatment of partial and complete septum (76.6% and 77.8% respectively), while residual septum had a live birth rate of 75.0%. This indicates that women with the most severe forms of uterine septa would still have a good chance of having successful live births, therefore justifying treatment for these women. The results also show that those women in the residual septum group who have undergone previous surgery for uterine septum have promising live birth rates, revealing that surgery should be repeated if necessary as it has shown the capability of producing favorable outcomes. There are not many studies in the literature comparing reproductive outcomes of women with complete and partial septa following hysteroscopic metroplasty, however a recent retrospective cohort study by Wang et al. reported statistically significant higher infertility rates after surgery for patients who had complete septum compared to those with partial septum (28.5% and 10.5% respectively) [22]. Similarly, an earlier study by Fedele et al. also reported lower cumulative pregnancy and live birth rates for women with partial septum compared to those with complete septum following hysteroscopic metroplasty [20]. However, another study by Paradisi et al. comparing reproductive outcomes for small and large partial septum (≤2.5 cm and >2.5 cm) found no statistically significant difference [23]. It appears that there is still a need for robust evidence to outline the depth of the septum that is clinically significant to have implications on the reproductive outcome and based on such evidence, potentially the current definitions of uterine septa could be revisited [16,24].

We found no statistically significant difference in the live birth rates when data was analyzed based on age. Despite this, live birth resulting from spontaneous conception was shown to be more common than IVF in women ≤39 years. IVF live births were more common for those who are older, likely due to the decrease in fertility potential with advancing age.

We did not routinely arrange follow up hysteroscopies or ultrasound examinations postoperatively, unless clinically indicated. In an earlier part of our practice we used to perform routine outpatient hysteroscopies postoperatively to check the cavity, this practice was later dropped to avoid unnecessary burdens on our patients and the health service, as we did not find any abnormalities at these follow up examinations.

One of the strengths of our study is that long term follow-up allowed for the identification of reproductive outcomes in the majority (~79%) of patients. Additionally, the same surgical technique was used consistently by the same surgeon who performed all the procedures, thus eliminating the variability seen in other published studies when more than one operator is involved especially of non-comparable experiences. Nevertheless, our study had some limitations. One of the limitations is that the retrospective nature of this study resulted in some missing data, however despite this it was possible to identify the reproductive outcomes in the majority of cases. Additionally, the sample size could have affected the reproductive outcome rates and statistical significance of results.

We could not address in this study whether or not hysteroscopic metroplasty results in better reproductive outcomes compared to expectant management due to the lack of a control group, which is another limitation of our study. Having a control group, would have provided a much clearer picture of how effective the treatment was. High quality evidence through randomized controlled prospective trials is still lacking on the efficacy of surgical treatment on improving the reproductive outcomes so as to generate evidence-based recommendations. However, based on retrospective observational studies assessing the pre- and post-operative reproductive outcome of women with septa and on non-randomized comparative studies between septum resection and expectant management, hysteroscopic septum treatment by experienced specialists in symptomatic women has been adopted as having a beneficial effect on reproductive outcome [7,8,10,11,13,14]. Similarly, the American Society for Reproductive Medicine (ASRM) and the National Institute for Health and Care Excellence (NICE), recommended hysteroscopic treatment of septate uterus in patients with infertility and unfavorable pregnancy outcomes [5,25].

This was questioned by a recent study by Rikken et al., comparing the surgical versus the expectant management. They reported that septum resection did not improve the reproductive outcomes and that compared to expectant management this led to a decreased chance of an ongoing pregnancy. They added that septum resection was not beneficial except for the possibility that surgery may lead to fewer cases of fetal malpresentation [15]. However, this study is severely criticized due to significant differences between the surgery and expectant management groups that led to bias. Saridogan et al., commented on the study that 36.8% of women managed expectantly had had at least one live birth before, compared to only 16.6% in the treatment group. In addition, 36.0% of women in the septum resection group were sub-fertile, as opposed to 20.4% in the expectant management group. This selection bias could potentially explain the finding of better reproductive outcomes in the expectant group [14]. Additionally, the diagnostic methods used varied significantly and technical details of surgery were missing in this retrospective study that involved 21 different centers over a period of 20 years [13,14,16,17].

There is universal agreement that prospective studies are still lacking to provide the required evidence needed to change our current practice that is the recommended by the literature. Until such evidence is available, we encourage other clinicians to publish their data in this field whilst prospective studies are planned. In addition, an international data registry could be beneficial to allow pooling of data in a standardized way.

## 5. Conclusions

Overall, encouraging reproductive outcomes were observed for women undergoing hysteroscopic surgery for uterine septum, in a patient population with a range of poor reproductive performance background. Similar high live birth rates were achieved in subgroups based on age, etiology/reason for referral and severity of uterine septum, suggesting surgery can be justified in all women seeking to conceive, including those presenting at an older age or with a more severe form of the condition. Therefore, until adequately powered multicenter randomized control studies assessing reproductive outcomes after hysteroscopic resection of uterine septum are available, our study results enhance the available evidence that hysteroscopic septum treatment has a beneficial effect in the accomplishment of a successful pregnancy outcome.

## Figures and Tables

**Figure 1 jcm-10-00130-f001:**
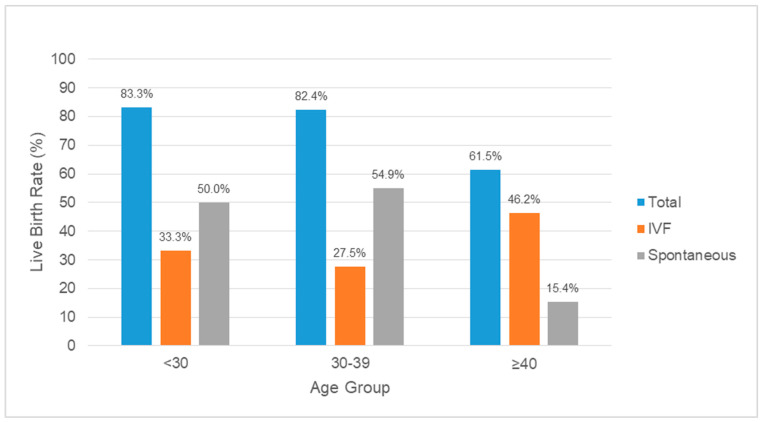
The live birth rates in different age groups following intrauterine septum treatment.

**Table 1 jcm-10-00130-t001:** Referral reasons of patients who received treatment for intrauterine septum.

Referral Reason (*n* = 99)	Number	%	95% CI
Infertility ^a^	54	54.5	44.7–64.4
Miscarriage (1 or 2)	22	22.2	14.0–30.4
Recurrent miscarriage ^b^	14	14.1	7.3–21.0
Pre-IVF surgery ^c,d^	7	7.1	2.0–12.1
Incidental finding	2	2.0	0–4.8
Total	99	100.0	

^a^ Three with miscarriages; ^b^ two with infertility; ^c^ one with recurrent implantation failure; ^d^ metroplasty prior to starting IVF for infertility due to other causes.

**Table 2 jcm-10-00130-t002:** Post-treatment reproductive outcomes of patients known to be trying for pregnancy.

Reproductive Outcome (*n* = 70)	Number	%
**Conceived spontaneously/with IUI**	**37**	**52.9**
Unknown Outcome	1	2.7
Miscarriage	3	8.1
Live birth	33	89.2
**Conceived with IVF**	**27**	**38.6**
Unknown Outcome	1	3.7
Miscarriage	3	11.1
Live birth	22	81.5
Termination of pregnancy	1	3.7
**Failed IVF**	**6**	**8.6**
**Total**	**70**	**100.0**

**Table 3 jcm-10-00130-t003:** The live birth rates based on the referral reasons of patients.

Referral Reason (*n* = 68)	Number Trying	Live Birth	% Live Birth (95% CI)
Infertility	38	30	78.9 (66.0–91.9)
Recurrent Miscarriage	10	8	80.0 (55.2–100)
Miscarriage	14	12	85.7 (67.4–100)
Pre-IVF Surgery	7	4	57.1 (20.5–93.8)
Incidental Finding	1	1	100.0

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
