# Peer review of "Reproductive Performance Following Hysteroscopic Surgery for Uterine Septum: Results from a Single Surgeon Data"

_jcm, 2021, doi:10.3390/jcm10010130_

Round 1
Reviewer 1 Report
As the present study involves data collected from patients, normally ethical approval by the authority responsible would be expected. If data was collected from a database already available and patients gave their consent for their data to be collected and analyzed later the shortcoming of lack of specific ethical approval for this study could be mitigated. It is unclear from the article if data on reproductive outcome has been collected unrelated from this study and thererfore being available already for analysis in this study or if patients were contacted specifically for this study to report on their reproductive outcome. In this case approval by the patient to be contacted would have to have been obtained beforehand. Clarification of this matter needs to be included in material and methods section.
Author Response
The study was assessed by the NHS Health Research Authority ‘Defining Research’ decision tool and a full ethical review by a NHS Research Ethics Committee or NHS/Health and Social Care Research and Development office was not required as this study was considered a service evaluation (www.hra-decisiontools.org.uk/research/). This has now been made clearer in the under 'Ethics Approval section.
As we already stated in the Methodology section the data was procured from existing data (patient records and clinic letters).
Reviewer 2 Report
The Authors present a retrospective analysis on 99 women who underwent hysteroscopic surgery for uterine septum between 16 January 2001 and December 2019. The results showed that surgical treatment of uterine septa was beneficial in improving reproductive outcome.
I read with great interest this paper. The manuscript is well written and tries to add new knowledge about the hot topic of uterine malformation treatment and infertility. Taking in consideration the large time period investigated, I hoped to find something new which helps the scientific community to build more robust evidences about this issue. Unfortunately, there are several bias which limit the quality of this paper.
Although the outcomes reported show a significant improvement in terms of live birth rate, the methodology described to make the diagnosis of uterine malformation represent a great bias which inevitably influence the quality of data and final results. Moreover, a detailed description of the class of uterine septum treated is completely lacking.
Regarding the effectiveness of the surgical treatment performed, the Authors did not report the methodology utilized for the follow up (by ultrasound? Hysteroscopy), and this represent an other important bias.
Line 74-77: the terms “mini-hysteroscope” is not appropriated. I suggest the Authors to better described the instrumentations utilized to perform metroplasties.
Line 77: The Authors reported an ultrasound guidance. The technique should be better explained (2d or 3d? which is the limit observed?).
Line 83: The authors should describe the methodology by which the follow up was performed after the surgical treatment.
Table 1: What do the Authors mean with “pre-IVF surgery. It is quite strange that no one among the patients studied would presented more than one indication (i.e. miscarriage and infertility).
I am seriously concerned about the reliability of uterine malformation diagnosis and this issue could represent a great bias which influences the quality of data reported.
The great part of the population studied had a diagnosis of uterine septum made by 2d ultrasound. Only in 5 cases the diagnosis was made after a hysteroscopy. No one was investigated by hysteroscopy in combination with 3d ultrasound. In 2 cases the method of diagnosis is unknow.
I realize that the large time period taken in consideration could be the explanation of the scarce application of 3d ultrasound, but I cannot understand how was possible to make a uterine malformation diagnosis without a hysteroscopy investigation.
Line 102: “preoperative diagnosis was made using ultrasound (with or without saline infusion) in 80 of 99” how was possible to perform a uterine malformation diagnosis by a 2d ultrasound with or without saline infusion?
Table 2: what does “top” mean?
The discussion section is definitely too long and should go to the point. In particular, the Authors should underline the limits of this study which seem to be more than those declar
Author Response
I read with great interest this paper. The manuscript is well written and tries to add new knowledge about the hot topic of uterine malformation treatment and infertility. Taking in consideration the large time period investigated, I hoped to find something new which helps the scientific community to build more robust evidences about this issue. Unfortunately, there are several bias which limit the quality of this paper.
Thank you. We recognise that this is a retrospective data analysis and that there is no control group. However these limitations have been discussed in the discussion section (lines 218-221).
Although the outcomes reported show a significant improvement in terms of live birth rate, the methodology described to make the diagnosis of uterine malformation represent a great bias which inevitably influence the quality of data and final results. Moreover, a detailed description of the class of uterine septum treated is completely lacking.
I think the reviewer may have missed the description of diagnosis in the Methodology section (lines 69-73). The diagnosis was usually made by 3D ultrasound as explained in the text, not by 2D. The technical details described in the article by Salim et al (2007) are by 3D ultrasound. The description of complete and partial septum has now been included in the text (lines 70-72). When the septum reached the internal os it was described as a complete septum, and a partial septum (or subseptate uterus) was diagnosed when the septum did not reach the internal os.
Regarding the effectiveness of the surgical treatment performed, the Authors did not report the methodology utilized for the follow up (by ultrasound? Hysteroscopy), and this represent an other important bias.
We did not routinely perform follow up hysteroscopies or ultrasound examinations to check the uterine cavity as explained in lines 86-87 and allowed patients to start trying for a pregnancy naturally or via fertility treatment. Further assessments were performed if there was a delay in getting pregnant or another miscarriage occurred. In fact, in earlier part of our practice we used to perform routine outpatient hysteroscopies postoperatively to check the cavity, this practice was later dropped to avoid unnecessary burden on our patients and the health service, as we did not find any abnormalities at these follow up examinations.
Line 74-77: the terms “mini-hysteroscope” is not appropriated. I suggest the Authors to better described the instrumentations utilized to perform metroplasties.
Further details of the hysteroscopes we use have been added in the Methodology section (lines 78-80). Alphascopes from Johnson and Johnson, or Bettochi operating hysteroscope, Karl Storz) with inflow, outflow and 5 or 7 French operating channels were used. Their diameters have been added in the text.
Line 77: The Authors reported an ultrasound guidance. The technique should be better explained (2d or 3d? which is the limit observed?).
2D ultrasound guidance was performed and a fundal myometrial thickness of 11-12 mm was aimed when the procedure was terminated.
Line 83: The authors should describe the methodology by which the follow up was performed after the surgical treatment.
Follow up was arranged on the basis of the clinical need, i.e patients who conceived attended antenatal care, those who needed fertility treatment attended fertility clinics, those with history of recurrent miscarriage attended recurrent miscarriage clinics, etc.
Table 1: What do the Authors mean with “pre-IVF surgery. It is quite strange that no one among the patients studied would presented more than one indication (i.e. miscarriage and infertility).
As explained at the bottom of the table there are patients with both miscarriage and infertility, and the numbers are already provided. 'Pre-IVF surgery' refers to those women who opted to have their septum treated prior to starting IVF treatment. This has now been to he bottom of the table.
I am seriously concerned about the reliability of uterine malformation diagnosis and this issue could represent a great bias which influences the quality of data reported.
The great part of the population studied had a diagnosis of uterine septum made by 2d ultrasound. Only in 5 cases the diagnosis was made after a hysteroscopy. No one was investigated by hysteroscopy in combination with 3d ultrasound. In 2 cases the method of diagnosis is unknow.
I realize that the large time period taken in consideration could be the explanation of the scarce application of 3d ultrasound, but I cannot understand how was possible to make a uterine malformation diagnosis without a hysteroscopy investigation.
Line 102: “preoperative diagnosis was made using ultrasound (with or without saline infusion) in 80 of 99” how was possible to perform a uterine malformation diagnosis by a 2d ultrasound with or without saline infusion?
As we explained above, the reviewer must have missed the diagnostic method in the Methodology section. We did not use 2D ultrasound for diagnosis, 3D ultrasound was the usual method of diagnosis.
Table 2: what does “top” mean?
top: termination of pregnancy. This has now been added to the table
The discussion section is definitely too long and should go to the point. In particular, the Authors should underline the limits of this study which seem to be more than those declar
The limitations of the study are already in the Discussion section. Some 'biases' the reviewer refers to are misunderstandings as explained above. These have been clarified in our response.

Round 2
Reviewer 2 Report
In fact, in earlier part of our practice we used to perform routine outpatient hysteroscopies postoperatively to check the cavity, this practice was later dropped to avoid unnecessary burden on our patients and the health service, as we did not find any abnormalities at these follow up examinations.
I suggest the Authors to add this matter among the limits of the study in the discussion section.
The limitations of the study are already in the Discussion section. Some 'biases' the reviewer refers to are misunderstandings as explained above. These have been clarified in our response.
The explanations made by the Authors were more than sufficient. Nevertheless, I would suggest the Authors to shrink the discussion section.
Author Response
Thank you for sending us the comments of Reviewer 2 on our revised version. We have now added a paragraph on follow up hysteroscopy/ultrasound examination in the Discussion section and reduced the length of Discussion as suggested by the reviewer. We hope the manuscript is now acceptable for publication.